# Tracking Temporal Development of Optical Thickness of Hydrogen Alpha Spectral Radiation in a Laser-Induced Plasma

**David M. Surmick [1]** and **Christian G. Parigger [2],\***

[1]  Physics and Applied Physics Department, University of Massachusetts Lowell, Lowell, MA 01854, USA; david_surmick@uml.edu
[2]  Physics Department, University of Tennessee Space Institute, Tullahoma, TN 37388-9700, USA
\*  Correspondence: cparigge@tennessee.edu; Tel.: +1-(931)-841-5690

**Abstract:** In this paper, we consider the temporal development of the optical density of the $H_\alpha$ spectral line in a hydrogen laser-induced plasma. This is achieved by using the so-called duplication method in which the spectral line is re-imaged onto itself and the ratio of the spectral line with it duplication is taken to its measurement without the duplication. We asses the temporal development of the self-absorption of the $H_\alpha$ line by tracking the decay of duplication ratio from its ideal value of 2. We show that when 20% loss is considered along the duplication optical path length, the ratio is 1.8 and decays to a value of 1.25 indicating an optically thin plasma grows in optical density to an optical depth of 1.16 by 400 ns in the plasma decay for plasma initiation conditions using Nd:YAG laser radiation at 120 mJ per pulse in a $1.11 \times 10^5$ Pa hydrogen/nitrogen gas mixture environment. We also go on to correct the $H_\alpha$ line profiles for the self-absorption impact using two methods. We show that a method in which the optical depth is directly calculated from the duplication ratio is equivalent to standard methods of self-absorption correction when only relative corrections to spectral emissions are needed.

**Keywords:** atomic spectroscopy; radiation transfer; hydrogen; laser-induced breakdown spectroscopy; stark broadening

## 1. Introduction

The act of tightly focussing a laser beam of sufficient energy creates a dynamic, micro sized plasma. The temperature and density properties of the decaying plasma depend on the laser, focal, and ambient conditions used at the onset of the plasma. For a nominal nanosecond pulsed laser with 10–100 mJ of energy per pulse, this plasma can have a temperature range of 0.5 to 5 eV and an electron density range between $10^{15}$ and $10^{19}$ cm$^{-3}$ depending on the plasma decay conditions and the laser ablation target [1]. Such plasma characteristics are ideal for use in nano-particle formation [2,3], pulsed laser deposition [4], and laser-induced breakdown spectroscopy (LIBS) [5–7]. In each of these applications optical spectroscopy becomes a primary tool (e.g., LIBS) for analysis and application, including use a tool for bench marking spectral line shapes with corresponding plasma conditions [8–10].

When the laser produced plasma is created, the hot and dense plasma state begins to cool and thermally expand. As this happens a cooler outer region of the plasma forms exterior to a hot plasma kernel. Radiation from this kernel is emitted through the plasma inner and outer regions along a particular line of sight. Along this line of sight, extinction of the radiation may occur and a spectroscopic line profile may become distorted resulting in absorption of the line [11]. For self-absorption, radiation is emitted by a specific transition in the hot plasma core and is absorbed to the same transition existing

in the region of the plasma. In the case of a typical LIBS plasma, the line shape is typically given as a Lorentzian line profile due to the Strong Stark broadening for these density ranges [12]. Alternatively, one can also model the line profile as a Voigt profile to include line shape contributions from the experimental apparatus and the relatively weak Doppler widths of spectral lines emitted in plasma with a temperature of the order of a few eV.

The impact of the absorption distortion on the line profile manifests in different ways depending on the amount of absorption. The tendency is for the line peak to flatten and broaden indicated by an increasing line width ($\Delta\lambda$) [13]. As the absorption becomes more extreme, the peak will take on a saturated form with a clear flat top peak and as the absorption becomes even more extreme the line will take on a reversed shaped with a clear central dip at the line center. This trend occurs for increasing optical depth $\tau$ and is displayed in Figure 1 for optical densities between 0 and 3. As can be seen, for moderate absorptions the line profile may not appear distorted (see $\tau = 0.5$ to 1). For applications such as quantitative LIBS where the line shape is pivotal for determining the plasma temperature and electron density [5], as well as determining elemental compositions from physical population [14] and univariate/multivariate models [5,15], distortions of the line profile need to be thoroughly addressed. Specific to the LIBS field of study, there is a long history of accounting for self-absorption [16–24]. Some methods depend on solving the equation of radiation transport [13,25–27], while others seek experimental corrections to specific line shapes of interest [23].

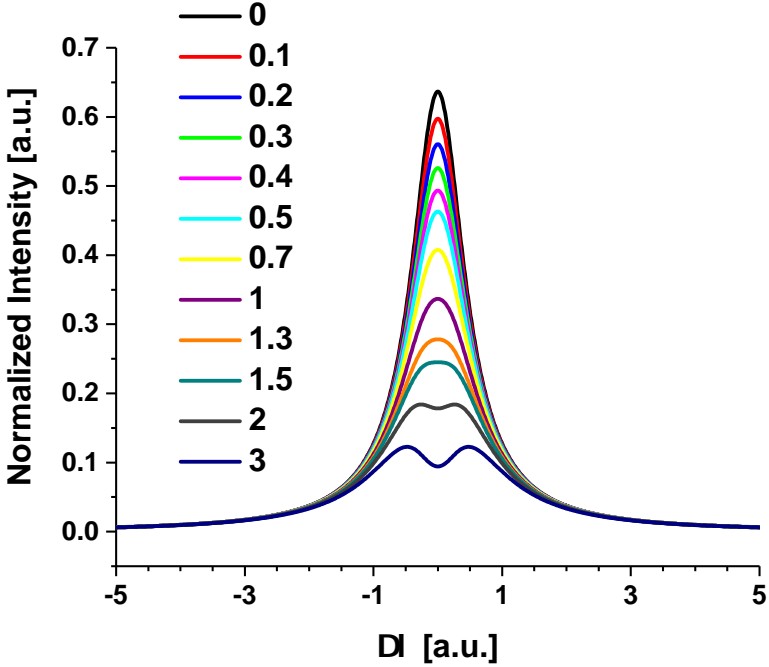

**Figure 1.** Alterations of a Lorentzian line profile under the influence of self-absorption for increasing optical density. The undistorted line width is 1 [a.u.].

In the present work, we detail the optical density of a laser-induced plasma throughout the plasma decay by monitoring the temporal progression of the hydrogen Balmer series $\alpha$ line ($H_\alpha$). This is completed through the standard practice of using a doubling mirror to re-image the plasma onto itself and take ratios of the plasma spectroscopic image collected with and without its doubled image. In the following section, we detail the standard method for applying the doubling mirror method and suggest a potential alternative that requires less post processing of the measured spectra. We go on to apply both methods to measurements of the $H_\alpha$ line and use the corresponding electron density determined from the line to indicate the usefulness of both methods of correcting the opacity of the spectral line.

## 2. Theory

The radiation transport equation details how light can be absorbed as it passes through a dense medium [11]. Specifically the amount of radiation ($L(\lambda, x)$) that leaves a column of absorbing material is detailed as

$$L(\lambda, x) = \epsilon(\lambda, x)dx - \kappa(\lambda, x)Ldx, \tag{1}$$

where $\epsilon(\lambda, x)$ the emission coefficient, $\kappa(\lambda, x)$ is the absorption along $dx$, and $dx$ is a slab of absorbing material. A solution to the radiation transport equation when spatial homogeneity is assumed is given by

$$L(\lambda) = S(\lambda)(1 - e^{-\tau(\lambda)})\mathcal{L}(\lambda), \tag{2}$$

where $S(\lambda)$ is the source function, which is taken as the ratio of the emission and absorption coefficients ($\epsilon(\lambda)/\kappa(\lambda)$). $\mathcal{L}(\lambda)$ is the normalized line profile and $\tau$ is the optical depth, which is defined as

$$\tau(\lambda) = \int_0^\ell \kappa(\lambda)dx, \tag{3}$$

where $\ell$ is the size of the absorption path length. When the source is in local thermodynamic equilibrium the source function takes the form of the Planck function [28]. In order to account for absorption, one can rearrange Equation (2) to isolate the emission coefficient as

$$\epsilon(\lambda)\mathcal{L}(\lambda) = L(\lambda)\frac{\kappa(\lambda)}{1 - e^{-\tau(\lambda)}} \tag{4}$$

such that if one can calculate a correction factor, apart from a multiplication of $\ell$, of the form

$$K(\lambda, \tau) = \frac{\tau}{1 - e^{-\tau(\lambda)}}, \tag{5}$$

absorption can be taken into account along a particular line of sight in a relative manner.

One method for taking absorption into account is to duplicate the emission source with reflective optics and compare the original emission source to the source with its duplication. In the case of laser-induced plasma, duplication is typically done using a retro-reflecting mirror [8,22,23]. The retro-reflection is reduced by a factor of $e^{-\tau}$ as it passes through the original emission source such that the ratio of the reflection plus the original source to the original emission is

$$R(\lambda) = \frac{S(\lambda)(1 - e^{-\tau}) + S(\lambda)(1 - e^{-\tau}G(\lambda)e^{-\tau})}{S(\lambda)(1 - e^{-\tau})}, \tag{6}$$

which reduces to

$$R(\lambda) = 1 + G(\lambda)e^{-\tau}, \tag{7}$$

where $G(\lambda)$ is a term that accounts for losses along the duplication optical path length. The method for tabulating a correction of the form of Equation (5) as outlined by Moon et al. [23] details using the ratio of the continuum with and without the emission duplication as a way of avoiding determining the optical losses, $G(\lambda)$. This method will hereafter be referred to the Kcorr method. In this scheme the correction factor is calculated as

$$K = \frac{\ln(y)}{y - 1}, \tag{8}$$

with

$$y = \frac{R_c(\lambda) - 1}{R(\lambda) - 1}, \tag{9}$$

where $R_c(\lambda)$ is the ratio of the continuum with and without the emission duplication. Rearranging Equation (7) shows that the loss factor shifts the value of the $\tau$ that can be determined from direct

division of the measured line profiles with and without the duplication mirror. Namely, for larger losses, smaller $\tau$'s and spectral intesnites are predicted.

As an alternative, we suggest directly finding $\tau(\lambda)$ from Equation (7) as a simpler method of calculating $K(\lambda, \tau)$ in Equation (5). This removes the need to find the ratio of the line continuum and also removes the need to differentiate this continuum from the line spectrum. This alternative method does however require one to provide an estimation of the losses along the optical path length of the duplication imaging system. In the present work we will show that in a relative sense, the need for this estimation has little impact for relative spectroscopic measurements and analyses, provided all lines used for analysis experience the same correction. The method will hereafter be referred to as the direct method of self-absorption correction.

## 3. Experimental Details

Laser-induced plasma was studied spectroscopically following plasma initiation from focused 1064 nm, Nd:YAG laser radiation. Self-absorption effects are studied by re-imaging the plasma onto itself prior to spectroscopic imaging through use of a plane, reflecting mirror. The plasma was initiated by focusing 120 mJ, 14 ns pulsed laser radiation through the window of a gas cell chamber. A 125-mm focal length UV-fused silica (UV-fs) plano-convex lens was used. The breakdown event was imaged onto the slit of a 0.64-meter Jobin-Yvon, HR640 Czerny Turner spectrometer installed with a 1200 grooves/mm grating. The gas cell was filled with 90% ultra-high purity (UHP, 99.999% pure) hydrogen gas and 10% UHP nitrogen gas and was evacuated with a mechanical/diffusion pump system to a pressure of $10^{-3}$ Pa ($1 \times 10^{-5}$ Torr) prior to filling the chamber volume with the desired gas mixture atmospheres. The chamber pressure at the time of the experiment was $1.11 \times 10^5$ Pa (838 Torr).

The plasma emissions were recorded with an Andor iStar intensified charge couple device (ICCD). The detector was a rectangular array of pixels. When coupled with the spectrometer, the horizontal arrays of pixels are used to record spectrally resolved data. The vertical pixels on the detector recorded spatially resolved data along the height of the spectrometer slit. The laser was focussed vertically through the top of the chamber. The size of the ICCD pixels were $13.6 \times 13.6$ μm. This resulted gave a spectral instrument resolution of approximately 0.15 nm for the selected slit width of 50 microns. Groups of 8 vertical rows were data-binned to achieve a spatial instrument resolution of 0.108 mm along the slit height. The imaging characteristics of the system were such that the plasma imaged onto the ICCD at a magnification of 1.05:1. The spectral and spatial resolutions (or spectral and spatial instrument resolutions) are the important figures of merit, and for completeness we included the actual slit width that was used. The spectral resolution is determined by the slit width, the spatial resolution by the binning. Both of which are affected by the modulation transfer function, especially of the intensifier. The instrument resolution is determined by analyzing the line shapes of low temperature, low density stand lamp sources used for wavelength calibration.

Self-absorption was studied using a plane, reflecting mirror and a lens to reflect the plasma onto itself prior to spectroscopic imaging. The plasma was first imaged onto the plane reflecting mirror. This image was then passed back through the lens and onto the plasma. The plasma and its duplicate were then imaged onto the spectrometer slit as is usually done in a typical LIBS experiment [29]. A block diagram of the apparatus is shown in Figure 2. The lenses used to image the plasma onto the mirror and the plasma and its duplicate image onto the spectrometer slit were identical and consisted of uncoated, UV-fs singlets with a focal length of 100 mm. To aid in the alignment of the apparatus, the mirror and the lens used to image the plasma onto the mirror were positioned with 5-axis Gimbal mounts for fine adjustments.

The alignment of the self-absorption apparatus is a delicate and vital component of this work given the spatial resolution of the ICCD. To aid in this procedure, the lens used to image the plasma with its duplication was fixed in its position so that it properly produced a focused image on the spectrometer slit. This corresponded to the location equivalent to the distance of 2× the focal length of the lens in relation to the spectrometer slit and 2× the back focal length (93.7 mm) in relation to

the breakdown plasma, when taking into account the thick-lens approximation. Likewise, the lens used to image the plasma onto the plane mirror was initially placed at a distance of twice the focal length of the lens in relation to the mirror and twice the back focal length of the lens in relation to the breakdown plasma. The best possible alignment was achieved by fine adjustments of both lenses and the mirror. The alignment was checked using zero order imaging with the spectrometer. A sufficient alignment was considered to be one in which ICCD images collected with and without the duplication were nearly identical. Further fine adjustments were made by comparing collected ICCD spectra with and without the duplication such that the ratio was as large as possible, with an ideal ratio limit of 2. These adjustments were made by viewing the Balmer series hydrogen beta line, $H_\beta$, at a time delay of 10 µs in SATP laboratory air breakdown. For this time delay, self-absorption for this spectral line is likely to be insignificant, especially in ambient air (as differentiated from the high pressure atmosphere used here) [30,31].

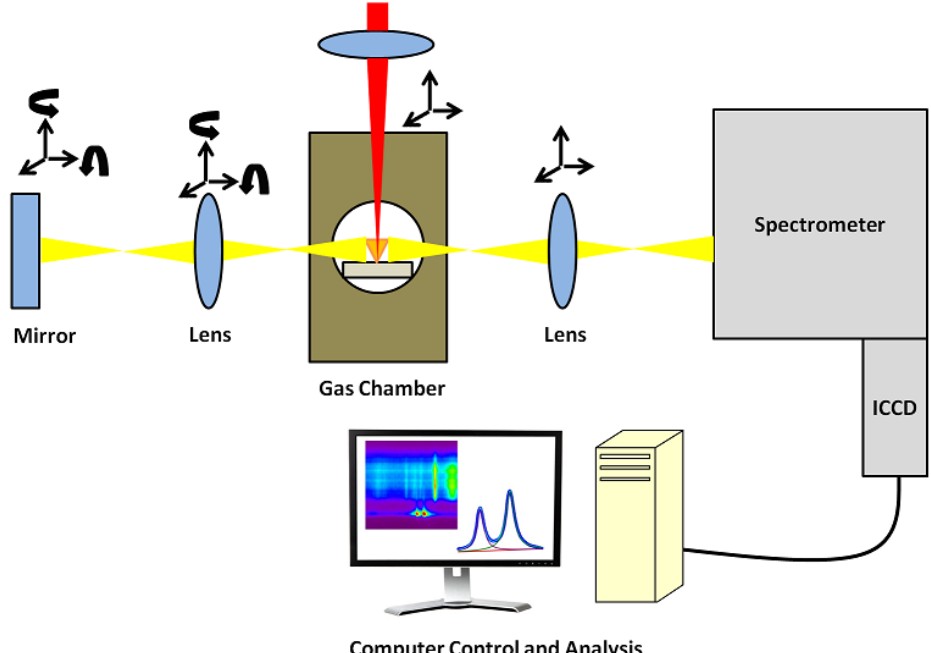

**Figure 2.** Block diagram of the experimental apparatus detailing self-absorption re-imaging.

$H_\alpha$ spectra were recorded at systematically varied time delays starting from 10 ns following plasma initiation up to a delay of 2150 ns. The selected gate width for the first 100 ns was 5 ns and a 150 ns gate width was used for all later measurements. Temporal resolution was achieved by synchronizing the ICCD to the Q-Switch output of the laser. Though spatial resolution was used for the initial measurement, all the $H_\alpha$ spectroscopic images were averaged after data collection to improve the signal quality and mitigate the impacts of a slight misalignment on the scale of the 0.1-mm spatial resolution along the vertical axis. This averaging excluded regions above and below the $H_\alpha$ emission where only signal noise was recorded. Moreover, this averaging precludes the use of interesting analysis from Abel-inversion techniques for the extraction of axial and radial self-absorption information. This is due to the sensitivity of the optical alignment in reference to our system's spatial resolution (0.108 mm) and not knowing the impact of the modulation transfer function (mapping of the source spatial extent to the image plane) in reference to our spatial resolution. Prior to analysis the spectrum was wavelength and intensity calibrated using a hydrogen glow discharge lamp and halogen light source, respectively.

## 4. Results and Discussion

### 4.1. Temporal Self-Absorption Behavior

Following collection and averaging of the $H_\alpha$ spectra both with and without the duplicating mirror, ratios of the doubled image to the non-doubled spectral image were calculated. The temporal development of the $H_\alpha$ spectra with and without the duplicating mirror is displayed in Figures 3 and 4 in the first 100 ns and at the later investigated times, respectively. The left image in each figure shows the spectra and the right image shows the spectra imaged with the duplication. These images show the rise in the intensity of the $H_\alpha$ line from the spectral continuum and its growth in intensity as the plasma cools and atomic recombination occurs. The spectra first become apparent after 30 ns as seen in Figure 3a. The $H_\alpha$ line is initially very broad but narrows gradually through time as the plasma decays. Prior to the 30-ns time delay, the plasma is characterized by strong continuum emissions. After reaching a peak intensity between 150 and 400 ns, the line begins to decay indicating the n = 3 levels of the hydrogen atoms are depopulating closer to the ground level as the plasma decays further and the plasma cools.

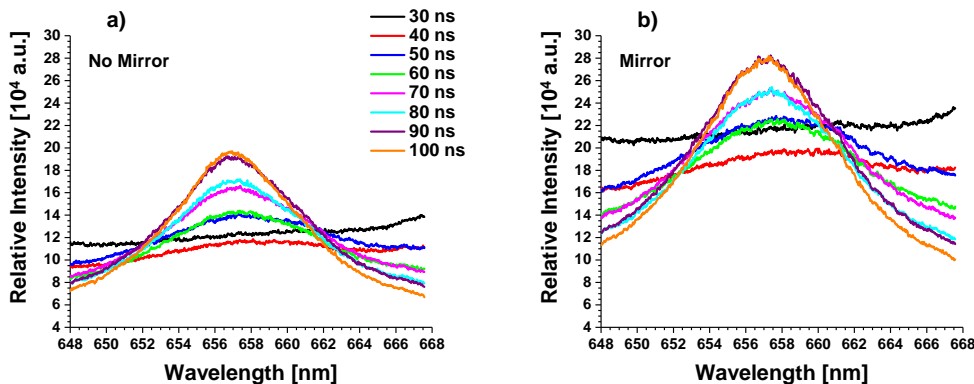

**Figure 3.** Temporal development of the $H_\alpha$ line in the first 100 ns following plasma initiation. (**a**) Measured spectra and (**b**) spectra measured with its duplication.

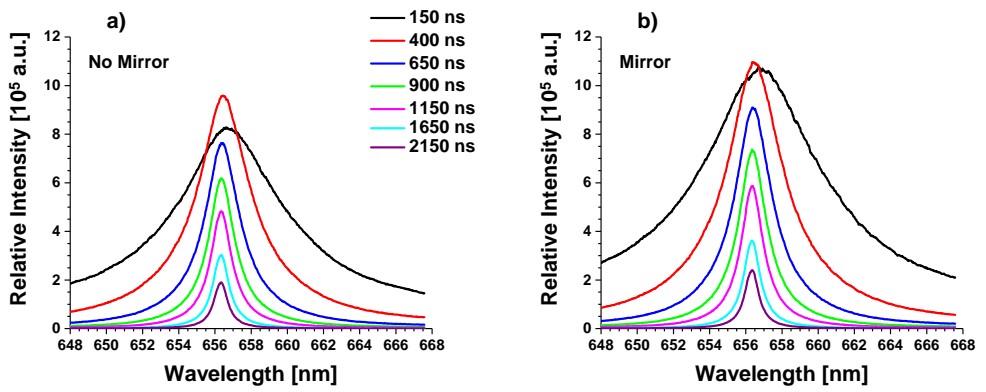

**Figure 4.** Temporal development of the $H_\alpha$ line between 150 ns and 2150 ns following plasma initiation. (**a**) Measured spectra and (**b**) spectra measured with its duplication.

To determine the temporal profile of the self-absorption of the $H_\alpha$ line, the measurements with and without the $H_\alpha$ duplication were used in conjunction with Equation (5) by two methods: (1) Equations (8) and (9) are used by finding the ratio of the spectra with and without its duplication and also finding the ratio of the continuum radiation for the Kcorr method and (2) by finding the optical depth according to Equation (7) and directly substituting this into the correction factor from

Equation (5) for the direct method. Both methods rely on finding the ratio of the spectra collected with and without its duplication. To gain insight into the temporal development of the self-absorption, these ratios have been tracked through the plasma decay and are displayed in Figure 5, showing times between 30 and 2150 ns.

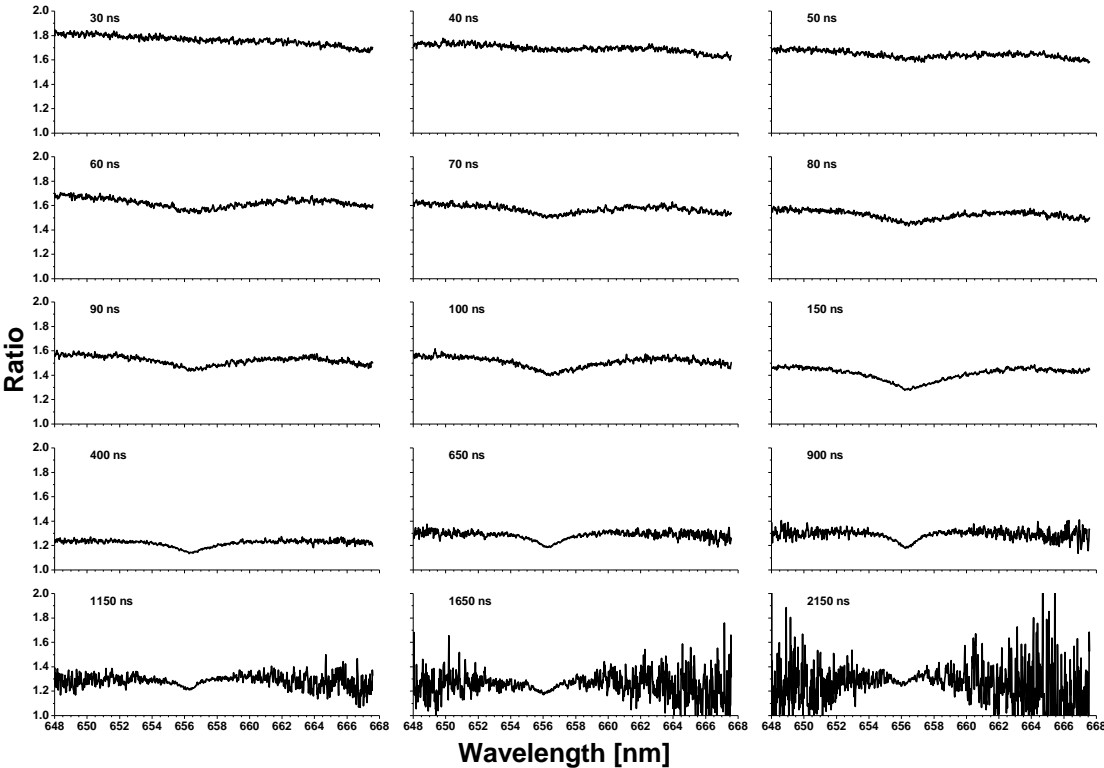

**Figure 5.** Ratios of the doubled $H_\alpha$ spectra to the non-doubled emissions through the plasma decay from 30 ns to 2150 ns. Each image has the wavelength and ratio ranges.

Initially, at 30 ns, the ratio is quite close to 2, the theoretical limit of the ratio. When losses are considered, a value of the approaching or exceeding 1.8 indicates little to no line self-absorption, when the signal almost entirely a continuum signal. This would be consistent with a plasma model in which the appropriate atomic levels of hydrogen (n = 2 and n = 3 for the Balmer $\alpha$ line) are just beginning to populate. As the plasma decay continues these levels become more populated, as evidenced by the rise in intensity of the $H_\alpha$ line. As this goes on, the ratio of the $H_\alpha$ line with and without its duplication begins to steadily decrease. By 100 ns the value of the ratio is approximately 1.6 and reaches a maximum low of about 1.3 to 1.25 at time later than 400 ns. After this time, there is an apparent rise the ratio indicating the amount of extinction along the optical line of sight is reducing. However, the plateau in the decay of the ratio more than likely indicates that a limit in the sensitivity of the technique has been reached.

Figure 5 also shows the tendency for the line center to be more perturbed by self-absorption, as was shown in Figure 1. As the line absorption becomes more prominent, a dip in the ratio begins to appear near the $H_\alpha$ line center. This is first seen in at a time of 60 ns and grows in prominence as the plasma decays. The magnitude of the dip relative to the mostly constant ratio in the line wings is between 0.05 to 0.1. Thus a minimum ratio of approximately 1.15 at the line center can be seen for the 400 ns time delay.

As the plasma decays, the line becomes more susceptible to experimental noise. The impact of this is amplified by dividing two spectra giving rise to the extreme noise seen in the ratios calculated after 800 ns. This noise addition can also be attributed to the narrowing nature of the $H_\alpha$ line as the wings of the later time ratios are most impacted. In this case the wings represent a part of

the spectrum that is characterized by an also decaying spectral continuum that will largely be the same between spectra collected with and without the duplication. Furthermore, this weakly intense continuum is more susceptible to noise contributions which is further amplified when the ratio is taken. This would also manifest in the early investigated times prior to 150 ns, the $H_\alpha$ line has contributions beyond the spectral range of our instrument, making it difficult to asses the contributions of the continuum radiation.

As a reference, Figure 6 shows the relationship between optical depth and the ratio of the spectral line with and without its duplication and without any losses given by Equation (5). At zero optical depth, the ratio is 2. As the optical depth increases to values greater than 1 ($\tau = 1$ is often cited as being optically thick [11]) the ratio approaches a limit of 1 indicating no radiation has emerged from the source. An optical depth of 1 corresponds to a ratio of nearly 1.37. Figure 5 suggests that a ratio of 1.25 in the line wings and 1.15 at line center can be extracted. This corresponds to optical depths of 1.38 and 1.89 at the line wings and line center, respectively, when no losses are considered. This is however an idealization that doesn't match the experimental conditions. If one were to assume a modest amount of loss along the duplication optical path length, such as 20% which means there is 93% transmittance all interfaces along duplication optical path, the optical depths become 1.16 and 1.67 at the line wings and line center, respectively. With a loss of 20% the ratio of 1.8 corresponds to an optically thin source. The increase in self-absorption as the plasma cools and decays has been reported in other studies as well [21,32,33].

Discussions of self-reabsorption [34] indicate that for an optically thin plasma and an ideal doubling mirror in place, the transmitted intensity amounts to twice that without the mirror. However, when the plasma is optically thick (at line center $\kappa_0 x \gg 1$) and for a Voigt parameter, $a$, or the ratio of Lorentzian and of Gaussian widths, $a \gg 1$, the line absorption, $A_L$, amounts to $A_L = 2 - \sqrt{2} = 0.59$. In other words, the transmission equals $\sqrt{2}$, i.e., 1.41 is the limiting value for the transmission. This results has been extensively communicated [35] and applied in the modeling of flame experiments [36] that utilize an experimental arrangement comprised of two consecutive line sources.

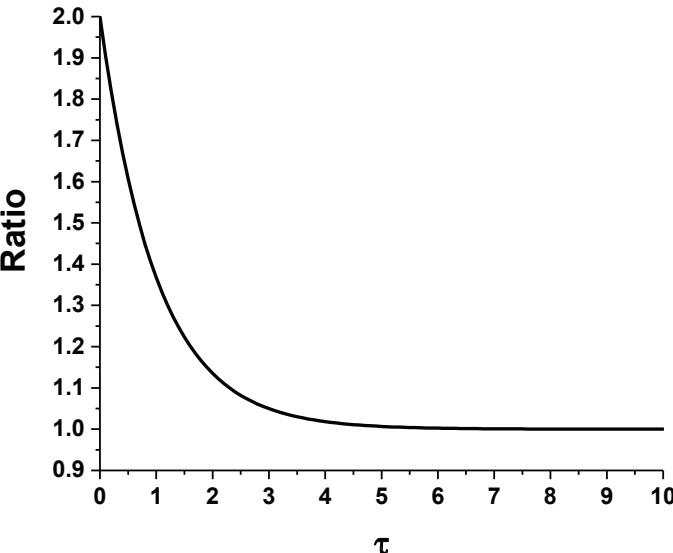

**Figure 6.** Relationship between optical depth, $\tau$, and the ratio of a spectral line with its duplication compared to the line without duplication.

## 4.2. Self-Absorption Impact on Line Shapes

The impact of the self-absorption on the temporal development is further discussed by applying the correction factor given in Equation (5) in two ways. The first is to use the Kcorr method outlined

by Moon et al. [23]. In performing this method, the ratios from Figure 5 are used but they have been smoothed with a second order Savitzky-Golay filter with a window size of 50 points along the spectral axis in order to reduce the impact of noise on the corrected spectral intensity vs wavelength. The spectral width of the filter is approximately 1 nm. While the 50-point, second order Savitzky-Golay filter is applied for determination of the correction factors, analysis of spectra may utilize a 5 to 10 point filter consistent with the system transfer function, i.e., the effective spectral resolution. However, Abel inversions are usually accomplished directly from the data without *a priori* smoothing when one employs a complete set of Chebyshev polynomials as elaborated in Ref. [37]. The ratio of the continuum was estimated by averaging the spectral end points on the desired range. These points were 645.5 and 667.547 nm. When the $H_\alpha$ line is sufficiently narrow ($\Delta\lambda = 4$ nm) these points indicate a distance from line center that is $5\times$ the line width, which provides a reasonable continuum estimation that is uninhibited by the $H_\alpha$ line profile. When the line is at its widest, these points are only $2\times$ the line width which introduces some uncertainty in determining the ratio of the continuum and also represents a limit of this method for narrow band spectra with broad spectral features.

The second method of correction is direct inversion of Equation (7) to find the optical depth using the ratios given in Figure 5 for the direct method. The ratios are once again smoothed using the Savitzky-Golay filter along the spectral axis. With this method one either needs to have an accurate characterization of the optical losses along the duplication optical path length or assume a reasonable loss. Characterizing the loss along duplication optical path length could be an arduous task that involves considering the modulation transfer function and other geometric optics impacts to determine the imaging system quality. At this stage, rather than take on this task, we instead consider three values of a loss to gauge the efficacy of using this method to correct the $H_\alpha$ spectra in a relative sense. The losses we consider include: No loss (an unlikely, ideal case), 10% loss, and 20% loss. For use in Equation (7), these values correspond to constant $G(\lambda)$ values of 1, 0.9, and 0.8. It is also reasonable to assume that the loss on a narrow spectral band of 22 nm is relatively constant.

Using three different values of the loss allows us to see how the line shape is impacted changing this parameter which will be important for further studies using this method of self-absorption correction. The no loss and 10% loss cases are likely ideal scenarios that would be difficult to experimentally achieve. The 20% loss case represents a best case scenario in which the optical system is perfectly aligned and highly efficient optics at the $H_\alpha$ wavelength are considered with a transmittance near 0.93 at each optical interface on the duplication optical path. After the loss is accounted for, the line profiles are now corrected using Equation (5) with the tabulated optical depths.

Figure 7 shows the result of applying both methods of correcting the $H_\alpha$ line at 100 and 600 ns time delays. For comparison, the measured line profile without its duplication is also presented. The most notable aspect of applying the two different methods is the Kcorr method produces a spectrum that is more comparable in intensity with the original line profile, however, the corrected line profile is now narrower (at both time delays) as would be expected. When the direct method as the self-absorption correction, the line intensity grows. The amount of intensity growth depends on the amount of loss. The greater the loss, the less the intensity of the line grows. The spectral features for each of the considered losses do appear to be narrower after correction, however.

Despite the line growth when the lines are corrected according to the direct method, the spectral intensities that are observed are relative in nature, as only a relative intensity correction of the experimental apparatus was performed. This is a common practice as absolute calibrations of spectral imaging systems are difficult to perform and are not necessary in most cases since quantitative information can be extracted from a relative intensity correction. For example, the line shape would only be impacted by an amplitude factor. Provided this amplitude factor is self consistent across an entire spectrum the quantitative analysis is not impacted. For the method at hand, provided it can be applied to multiple lines self consistently, further methods should not be impacted, though one should perform a baseline study for each new type of analysis that is considered (Boltzmann plots, multivariate modeling, etc.).

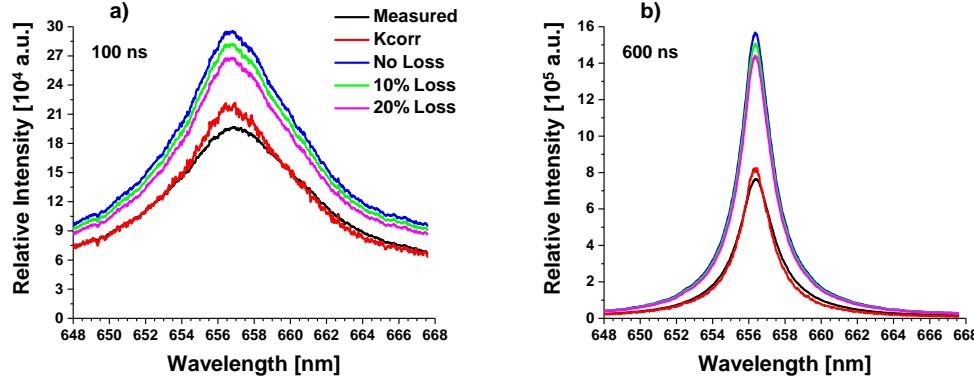

**Figure 7.** Results of correcting the $H_\alpha$ spectrum using the methods of Equations (8) and (9) and also directly calculating the optical depth with assumed losses in conjunction with Equation (7) for (**a**) 100 ns and (**b**) 400 ns time delays in the plasma decay.

To compare the two methods of self-absorption correction and verify the line profile remain unaltered regardless of the method, the uncorrected and corrected line profiles were fit in order to calculate the line width. The line width was used to calculate the electron density to see the impact on determining the plasma state. The line profiles were fit with a Voigt line profile, an integral line shape function resulting from the convolution of Lorentzian and Gaussian line shapes [11]. The Lorentzian component of the Voigt profile represents the Stark broadened $H_\alpha$ line, while the Gaussian component represents in the instrument function width of 0.15 nm (determined during spectrometer-detector calibration) under the assumption of negligible Doppler broadening. One should note that the exact line shape of a Stark broadened hydrogen Balmer series line is a Holtsmarkian line profile, however for ease of analysis the Lorentzian line profile is an acceptable approximation [38–40].

The integral Voit line profile is numerical calculated using Fadeeva function or also called complex complimentary error function with the fixed Gaussian width of 0.15 nm [41,42]. The line profile fitting was implemented in the MatLab® scripting environment [43]. The Fadeeva function was numerically calculated using the Algorithm 916 method of Zaghloul et al. [44]. Line shape fitting was carried using a Trust Region nonlinear curve fitting routine [45,46] in which the fit parameters were the line width, amplitude, and shift as well as two terms used characterize a linear offset to model any continuum components in the spectra signals.

In terms of characterizing the electron density of the plasma, one typically uses the line width. As a result, the behavior of the width before and after the self-absorption corrections is considered. These results are displayed in Table 1 and detail the line width progression from 40 ns to 2150 ns. The 30 ns time is left out from this discussion because the $H_\alpha$ line has not yet emerged from the spectral continuum. The table shows the what the uncorrected line width would be and the corrected line widths using The Kcorr and direct methods with no, 10%, and 20% losses. The table shows that initially the $H_\alpha$ line is very broad ($\Delta\lambda > 9$ nm) and subsequently decays to a relatively narrow line ($\Delta\lambda < 1$ nm). The uncertainties on the indicated line widths represent contributions from both the minimum spectral resolution and uncertainties introduced during line profile fitting. These uncertainties are greatest when the line is at its broadest and decay to a minimum value of 0.15 nm representing little to no uncertainty contribution from the fitting algorithms. This occurs after 400 ns when the line width is less than 4 nm and 5× the line width occurs on the investigated spectral range, indicating the spectral continuum is likely to be well characterized within the measurement.

Table 1 shows that regardless of the method of correcting the spectrum, the line width is reduced when the self-absorption factor is applied. The reduction in line widths ranges from 9 to 3.5 percent from the early times to the later plasma decay times, respectively. Furthermore, the two methods of correcting for the self-absorption return similar values of the line width after correction. The amount of loss considered in the relative correction model also produces similar line widths. Early in the plasma

decay the Kcorr methods tends produce line widths that are 0.1 nm greater than the Equation (7) method. This can be attributed to the difficulty in appropriately finding the continuum ratio when the line is still very broad.

**Table 1.** Line widths of the $H_\alpha$ line prior to and after self-absorption correction using all the stated methods. The 30 ns time delay is excluded form the table because the $H_\alpha$ has not yet emerged from the continuum at this point.

| Time [ns] | Uncorrected $\Delta\lambda$ [nm] | Kcorr $\Delta\lambda$ [nm] | No Loss $\Delta\lambda$ [nm] | 10% Loss $\Delta\lambda$ [nm] | 20% $\Delta\lambda$ [nm] |
|---|---|---|---|---|---|
| 40 | $8.42 \pm 0.34$ | $7.33 \pm 0.43$ | $6.80 \pm 0.34$ | $6.77 \pm 0.34$ | $6.74 \pm 0.34$ |
| 50 | $9.80 \pm 0.28$ | $9.01 \pm 0.28$ | $8.89 \pm 0.25$ | $8.88 \pm 0.25$ | $8.86 \pm 0.25$ |
| 60 | $9.24 \pm 0.23$ | $8.32 \pm 0.23$ | $8.20 \pm 0.23$ | $8.19 \pm 0.23$ | $8.17 \pm 0.23$ |
| 70 | $8.98 \pm 0.26$ | $8.12 \pm 0.21$ | $8.03 \pm 0.21$ | $8.01 \pm 0.22$ | $7.99 \pm 0.21$ |
| 80 | $8.57 \pm 0.24$ | $7.63 \pm 0.19$ | $7.64 \pm 0.20$ | $7.61 \pm 0.20$ | $7.60 \pm 0.20$ |
| 90 | $8.11 \pm 0.21$ | $7.34 \pm 0.19$ | $7.31 \pm 0.19$ | $7.30 \pm 0.19$ | $7.29 \pm 0.19$ |
| 100 | $7.80 \pm 0.20$ | $6.93 \pm 0.18$ | $6.93 \pm 0.18$ | $6.92 \pm 0.18$ | $6.90 \pm 0.18$ |
| 150 | $6.30 \pm 0.17$ | $5.46 \pm 0.16$ | $5.51 \pm 0.16$ | $5.49 \pm 0.16$ | $5.47 \pm 0.16$ |
| 400 | $3.72 \pm 0.15$ | $3.17 \pm 0.15$ | $3.28 \pm 0.15$ | $3.28 \pm 0.15$ | $3.27 \pm 0.15$ |
| 650 | $2.53 \pm 0.15$ | $2.19 \pm 0.15$ | $2.27 \pm 0.15$ | $2.26 \pm 0.15$ | $2.25 \pm 0.15$ |
| 900 | $1.88 \pm 0.15$ | $1.58 \pm 0.15$ | $1.66 \pm 0.15$ | $1.66 \pm 0.15$ | $1.65 \pm 0.15$ |
| 1150 | $1.51 \pm 0.15$ | $1.36 \pm 0.15$ | $1.40 \pm 0.15$ | $1.40 \pm 0.15$ | $1.40 \pm 0.15$ |
| 1650 | $1.10 \pm 0.15$ | $1.02 \pm 0.15$ | $1.04 \pm 0.15$ | $1.04 \pm 0.15$ | $1.04 \pm 0.15$ |
| 2150 | $0.90 \pm 0.15$ | $0.57 \pm 0.15$ | $0.87 \pm 0.15$ | $0.87 \pm 0.15$ | $0.87 \pm 0.15$ |

When the uncertainties are considered, the line widths can be considered to be the same for both methods of self-absorption correction and all three losses considered. There are two exceptions when the Kcorr value does not match the direct method. At 40 ns, the Kcorr method is 0.5 nm greater, yet there is a slight overlap when the uncertainties are considered. At this time, the $H_\alpha$, while noticeable, is still emerging from the continuum and is also very broad making it difficult to ascertain the ratio of the continuum. This point was also the most uncertain in terms of the line profile fitting. The second point with a notable discrepancy is the 2150-ns time delay value for Kcorr. At this time the value of the ratio of the continuum is also difficult to ascertain due to the decaying nature of the plasma. As the plasma decays this value will tend toward unity which causes a singularity in the expression for Kcorr in Equation (8). This represents an advantage of the method using Equation (7) directly, as less processing of the spectra is required in order to obtain a correction factor. With the fact that the line profile is preserved in a relative sense one may prefer this method even though an estimate of the loss along the duplication optical path is required.

The final investigation of this work is to consider the electron density, $N_e$, decay of the plasma. This is shown in Figure 8 along with the line width decay of the $H_\alpha$ line. The $N_e$ was determined $H_\alpha$ line width using the empirical formula outlined in Reference [47]. For the $H_\alpha$ line, the $N_e$ is approximately proportional to $\Delta\lambda^{2/3}$ [38,39,47,48]. This figure mimics the results shown in Table 1. The line widths and electron densities are reduced when each of the self-absorption correction methods are applied and the each of the corrected densities and line widths agree with each other, especially when the uncertainty is considered. The impact of not accounting for self-absorption, even when moderate absorption is clear from Figure 8. The electron density can be over estimated. For quantitative analysis, such as calibration free LIBS, this can alter the elemental compositions that are determined, over and beyond any impact that the line shape distortion may cause. Also as expected, the plasma is seen decay in density as the plasma cools and atomic and molecular recombination occurs. The plasma is seen to decay from an electron density of approximately $2 \times 10^{18}$ cm$^{-3}$ at 50 ns to approximately $5 \times 10^{16}$ cm$^{-3}$ at 2150 ns, following some manner of exponential decay as one might expect.

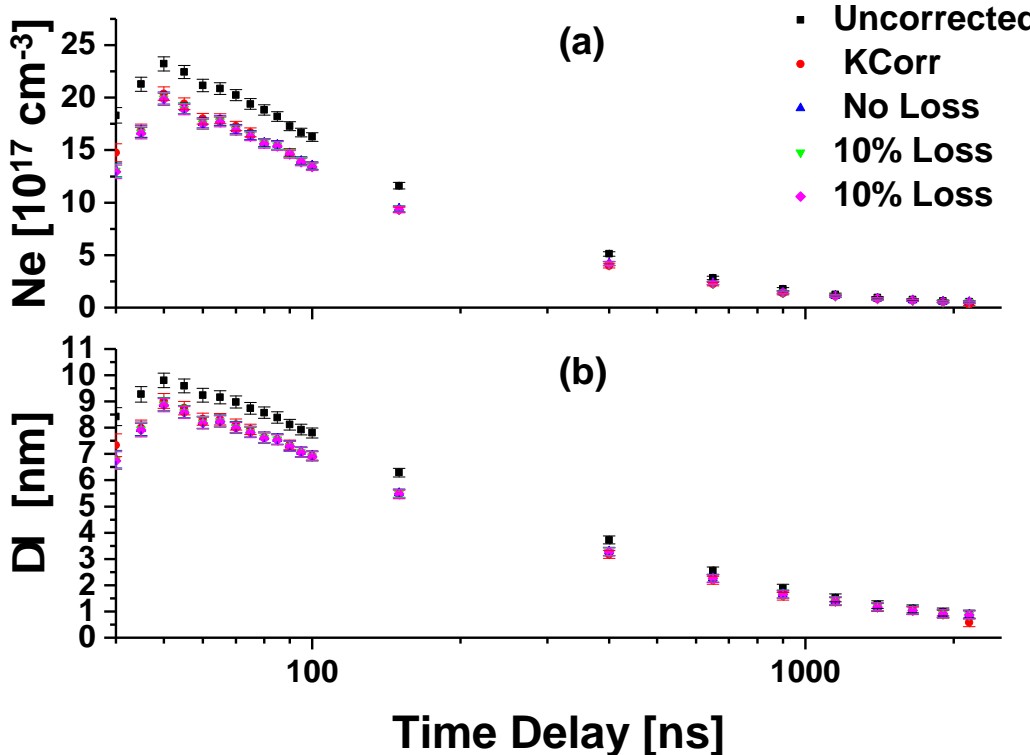

**Figure 8.** Decay of the plasma (**a**) electron density and (**b**) $H_\alpha$ line width through time.

## 5. Conclusions

In this work, the temporal development of the self-absorption of the $H_\alpha$ line was considered by studying the duplication method of re-imaging the spectral line onto itself and comparing the line with and without this duplication. The temporal development of this ratio indicated that the early stages of the plasma decay there is little self-absorption occurring in the plasma as the ratio value was 1.8. Throughout the plasma decay this value reduced to a minimum of 1.25 indicating an optical depth of 1.16 when a loss of a 20% is considered along the duplication optical path length. This result showed that as the plasma decayed and cooled the appropriate levels of hydrogen became sufficiently populated to induce self-absorption of the spectral line even if it is not readily obvious via visual inspection of the line profile.

The spectral line profiles were then corrected using two methods. The first was the standard method outlined by Moon et al. [23]. The second involved a direct calculation of the optical depth from the ratios of the spectra measured with and without its duplication. This required the assumption of some form of loss along the duplication optical path length. Investigation of the corrected profiles using line profile fitting methods with an emphasis on preserving the line width showed that both methods corrected the line self-absorption equally by finding the same line width within experimental uncertainty. The amount of loss considered along the duplication optical path only impacted the intensity of the signal, such that in an experiment where only relative intensities are required either method of self-absorption correction may be considered.

**Author Contributions:** All authors contributed equally to this work.

**Funding:** This research received no external funding.

**Acknowledgments:** The authors wish to acknowledge the support of the Center for Laser Applications at the University of Tennessee Space Institute for partial support of the experimental work of this effort.

**Conflicts of Interest:** The authors declare no conflict of interest.

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
