# Peer review of "Tracking Temporal Development of Optical Thickness of Hydrogen Alpha Spectral Radiation in a Laser-Induced Plasma"

_atoms, doi:10.3390/atoms7040101_

Round 1
Reviewer 1 Report
Dear authors,
The presented manuscript is sound, the thought flow is fluid and it could be easily followed and and understood. There is one question, is there an attempt to make some sort of Abel inversion of multiple averaged spectral images? Although the usage of second order Savitzky-Golay filter of window size of 50 bins is rather wide, maybe the Abel inversion on several shots could yield to some more applicable results.
Sincerely
Author Response
We kindly thank the time attention, comments, and support of Reviewers 1 and 2. Below we detail our responses to their comments. Changes to the manuscript in response to Reviewer 1 are marked with blue text in the revised manuscript. A revised copy of the manuscript is attached with the following details in response to each reviewer.
Reviewer 1:
The presented manuscript is sound, the thought flow is fluid and it could be easily followed and and understood. There is one question, is there an attempt to make some sort of Abel inversion of multiple averaged spectral images? Although the usage of second order Savitzky-Golay filter of window size of 50 bins is rather wide, maybe the Abel inversion on several shots could yield to some more applicable results.
The Savitzky-Golay filter is applied to smooth the intensity signal along the spectral axis for ratios as shown in Figure 5. All spatial components of the signal were removed by taking the spatial average of the spatially resolved spectral signal due to physical limitations during the optical path alignment. Though Abel inversions of axially resolved spectral measurements in reference to our self-absorption analysis procedure would be of future interest, it is not appropriate to attempt the spatial deconvolution with the data we present in the current manuscript. This is due to the difficulties in accurately aligning the optics to within uncertainty tolerances of the ICCD (<0.2 mm) and the difficulty in taking on the non-trivial task of detailed ray trace modeling (Zeemax) to determine how light propagates through our experimental apparatus to determine a modulation transfer function detailing spatial mapping of our emission source onto our image plane at our ICCD. These two points in conjunction with each other preclude us from considering Abel inversions for the presented self-absorption correction procedures.
We make clear that the Savtizky-Golay filter applies only to the spectral axis of the ratios on lines 236-242 in the revised manuscript with the following text:
“In performing this method, the ratios from Figure 5 are used but these ratios have been smoothed with a second order Savitzky-Golay filter with a window size of 50 points along the spectral axis in order to reduce the impact of noise on the corrected spectral intensity vs wavelength. The spectral width of the 50 point filter is approximately 1 nm. While the 50-point, second order Savitzky-Golay filter is applied for determination of the correction factors, analysis of spectra may utilize a 5 to 10 point filter consistent with the system transfer function, i.e., the effective spectral resolution. However, Abel inversions are usually accomplished directly from the data without a priori smoothing when one employs a complete set of Chebyshev polynomials as elaborated in Ref. [37]”
and also in line 247:
“The ratios are once again smoothed using the 50 point Savitzky-Golay filter along the spectral axis.”
We further expand our reasoning for taking the spatial average and precluding Abel inversion techniques as an analytical tool for the data we present in this manuscript on lines 158-162 in the revised manuscript with the following text:
“This averaging precludes the use of interesting analysis from Abel inversion techniques to extract axial and radial self-absorption information. This is due to the sensitivity of the optical alignment in reference to our system's spatial resolution (0.108 mm) and not knowing the impact of the modulation transfer function (mapping of the source spatial extent to the image plane) in reference to our spatial resolution.”
Reviewer 2 Report
An excellent experimental plasma spectroscopy method is presented and supported with the high quality measurements.
The paper presents two comparative methods of correction for the spectral line profile self-absorption in plasma.
The authors have evaluated the spectroscopic method that measures the spectral line profiles with and without line profile duplication (single and a double optical path through the plasma) to correct the observed line profile for self-absorption in plasma.
The experimental setup is described in details to familiarize the reader with the advantages of the proposed technique as well as the severity of self-absorption effects in plasma. The described method, based on the line profile with and without duplication is a useful alternative to the direct method. It requires only the relative measurements, similar to the direct method. The high quality of the measured data gives a confidence in the validity and accuracy of the proposed method.
I recommend this submission to be published in present form.
Author Response
We kindly thank the time attention, comments, and support of Reviewers 1 and 2. Below we detail our responses to their comments. Changes to the manuscript in response to Reviewer 1 are marked with blue text in the revised manuscript. A revised copy of the manuscript is attached with the following details in response to each reviewer.
Reviewer 2:
An excellent experimental plasma spectroscopy method is presented and supported with the high quality measurements.
The paper presents two comparative methods of correction for the spectral line profile self-absorption in plasma.
The authors have evaluated the spectroscopic method that measures the spectral line profiles with and without line profile duplication (single and a double optical path through the plasma) to correct the observed line profile for self-absorption in plasma.
The experimental setup is described in details to familiarize the reader with the advantages of the proposed technique as well as the severity of self-absorption effects in plasma. The described method, based on the line profile with and without duplication is a useful alternative to the direct method. It requires only the relative measurements, similar to the direct method. The high quality of the measured data gives a confidence in the validity and accuracy of the proposed method.
I recommend this submission to be published in present form
We graciously thank Reviewer 2 his/her kind words and support of our effort in the presented manuscript.